# Prevalence of Gastrointestinal Parasites in Small Ruminant Farms in Southern Spain

**DOI:** 10.3390/ani14111668

**Published:** 2024-06-03

**Authors:** Pablo José Rufino-Moya, Rafael Zafra Leva, Lilian Gonçalves Reis, Isabel Acosta García, Diego Ruiz Di Genova, Almudena Sánchez Gómez, Francisco García García, Francisco J. Martínez-Moreno

**Affiliations:** 1Animal Health Department (Parasitology and Parasitic Diseases), Faculty of Veterinary Medicine, University of Córdoba, Sanidad Animal Building, Rabanales Campus, 14014 Córdoba, Spain; pablo.rufino.moya@gmail.com (P.J.R.-M.); lilian.preis1@gmail.com (L.G.R.); sa1acgai@uco.es (I.A.G.); fjmartinez@uco.es (F.J.M.-M.); 2COVAP (Cooperativa Ganadera del Valle de los Pedroches) R&D Department, Pozoblanco, 14400 Córdoba, Spain; demruiz@gmail.com (D.R.D.G.); asanchezgo@covap.es (A.S.G.); fgarciag@covap.es (F.G.G.)

**Keywords:** small ruminants, survey, management, gastrointestinal parasites

## Abstract

**Simple Summary:**

This study investigates the prevalence of gastrointestinal parasites in small ruminant farms (159 sheep and 39 goats) in southern Spain, a key area for the country’s livestock production. This research reveals that *Eimeria* spp. Is the most common parasitic infection, followed by Strongyles. Other parasites, like *Moniezia* spp., *Trichuris* spp., and *D. dendriticum*, are less prevalent but more common in sheep. This study also includes a survey on farmers’ management practices, highlighting that regular monitoring through coprological analyses is not common; veterinarians are seldom involved in deworming plans, and anthelmintic treatment is typically applied twice a year in sheep and once in goats. This report suggests that implementing certain management measures could potentially mitigate parasite infections. This constitutes the first report on the epidemiological status of gastrointestinal parasites in small ruminants in southern Spain.

**Abstract:**

The primary population of small ruminants in Spain is concentrated in the southern region, a critical area for the country’s livestock production. Indirect economic losses can occur when this livestock is affected by gastrointestinal parasites. This study aimed to determine the prevalence of these parasites in small ruminant herds (159 sheep and 39 goats) through coprological analyses and conducted a survey on farmers’ management practices related to gastrointestinal parasite control. The survey results revealed some important aspects: monitoring through coprological analyses is not a common practice; veterinarians are not typically involved in deworming plans; anthelmintic treatment in adults is often applied twice a year in sheep and once a year in goats; and finally, drug rotation was higher in sheep farms. Coprological analyses showed *Eimeria* spp. as the most common parasitic infection, followed by Strongyles infection. Other parasites like *Moniezia* spp., *Trichuris* spp., and *D. dendriticum* were less important, although their prevalence was higher in sheep than goats. This constitutes the first report on the epidemiological status of gastrointestinal parasites in small ruminants in southern Spain. Based on the survey findings, the introduction of certain management measures on farms could potentially mitigate parasite infections.

## 1. Introduction

Small ruminant farming is a significant economic sector in Spain. In fact, within the European Union, Spain is home to 25% of the sheep population and 22% of the goat population. The economic benefits of their production in 2020 were estimated to be around EUR 2000 million [1].

However, parasitic infections, particularly those caused by gastrointestinal nematodes (GIN) and *Eimeria* spp., are the primary agents responsible for production decreases in many intensive and extensive farming systems. These parasitic infections lead to indirect economic losses related to a lower milk yield, decreased weight gain [2,3,4], increased feed demand [3,5], and medical treatment costs [6,7,8].

Additionally, there are direct economic losses related to mortalities when parasitation is more severe [9].

Control measures against gastrointestinal nematodes primarily rely on the use of anthelmintic drugs [8]. However, their widespread and uncontrolled use has led to the emergence of anthelmintic resistance (AR) [8,10]. In this regard, three factors have been identified as the leading causes of parasite survival following anthelmintic treatment [11]: (1) frequent treatments; (2) anthelmintics under-dosage; and (3) the lack of rotation of active principles [12,13,14]. These factors, along with certain types of farm management, can promote the development of AR, which requires higher doses of anthelmintics to ensure efficacy. This is particularly true for goats, as these ruminants metabolize and eliminate these drugs faster than sheep [15,16].

Previous studies carried out in Spain about management measures show that less than 50% of farmers apply quarantine to new animals in Aragón [17,18], Castilla-León, Castilla-La Mancha, Extremadura, and Andalucía [19]. Likewise, the most frequent deworming protocol includes two treatments per year in the aforementioned locations [17,18,19]. Nevertheless, in areas of Galicia, the protocol followed by more than 60% of farmers only performs one treatment per year [20]. Alternatively, the rotation of anthelmintic drugs is not very common, especially in Aragón, where 40–68% of the farms only apply one type of anthelmintic in herds [17,18,19]. Coprological analysis prior to deworming is not a common practice either. In Aragón, it is performed in less than 36% of farms; in Castilla-La Mancha, 2%; and in Galicia and Castilla-León, it is not performed at all [18,20]. The use of veterinary advice as a source of information for anthelmintic applications varies among these studies. While some authors describe veterinary participation in more than 90% of the farms [18,19], others found a participation rate of only 11% of the farms [20].

AR in sheep has been reported in various regions of Spain, with the level of resistance varying according to the active principle employed. In Castilla-León, AR has been reported against benzimidazoles (12.7–30%), macrocyclic lactones (15.7–27.3%), and levamisole (34.6–60%) [8,21,22]. Similarly, in Aragón, AR against benzimidazole was reported at 11% [17], while in Galicia, AR against benzimidazoles and macrocyclic lactones was reported at 18.1 and 2.8%, respectively [23].

In other countries such as Italy, resistance to benzimidazole and macrocyclic lactones has been found in 30–50% and 20–50% of the analyzed flocks, respectively [24,25]. In Poland, goat flocks have shown higher resistance to benzimidazoles (88%), macrocyclic lactones (95%), and levamisole (12%) [26].

In Spain, few studies have focused on AR, management practices, and the application of anthelmintics on small ruminant farms [18,19,20]. Excluding the study by Rojo-Vázquez and Hosking (2013), which incorporated some farms from southwestern Spain, the aforementioned studies were mainly conducted in central and northern Spain. However, most of the total sheep and goat population in Spain is predominantly located in two regions, Andalusia and the area encompassing Extremadura and Castilla La Mancha [1]. Survey results regarding farm management, parasite control, and the use of anthelmintics reveal significant variability in the data depending on the Spanish region studied [18,19]. Consequently, the results of these studies, which were primarily conducted in northern and central Spain, should not be extrapolated to the environmental and management conditions prevalent on farms in southern Spain, especially Andalusia. Therefore, it is crucial to understand the epidemiological situation and the management practices implemented by farmers in this region. It appears necessary to identify high-risk management practices that could potentially be associated with a high parasite burden, which may be linked to the occurrence of anthelmintic resistance.

Considering the above, this research aims to investigate the epidemiological situation of gastrointestinal parasites on farms in southern Spain through coprological studies. Additionally, a survey was conducted among farmers to assess the potential influence of management measures on the presence of a high parasite burden and anthelmintic resistance. To this end, 159 sheep farms and 39 goat farms from southern Spain were surveyed, and coprological analyses were performed.

## 2. Materials and Methods

### 2.1. Study Area and Farm Selection

The current study was performed in the provinces of Córdoba, Badajoz, and Ciudad Real in southern Spain. This region spans an area of 55,350 km^2^ and experiences a temperate climate with dry, hot summers, as per the Köppen–Geiger Climate Classification for the Iberian Peninsula. The area exhibits a mean annual temperature range of 12–17.5 °C and a mean annual rainfall range of 400–800 mm. The farms involved in this study are part of the company COVAP (Cooperativa Ganadera del Valle de los Pedroches). This research focused on 198 farms from April 2021 to May 2021.

### 2.2. Farm Questionnaire

This study was carried out in partnership with the company COVAP. All sheep and goat farms affiliated with this cooperative were invited to participate in the study, resulting in the analysis of 159 sheep (average size of 599.83 (range 30–4064)) and 39 goat farms (average size of 309.82 animals (range 125–600)). Farmers were approached by their veterinarians to complete a questionnaire, thereby gathering information related to farm management and practices for controlling gastrointestinal parasites.

The survey comprised twenty-one closed questions, categorized into three sections (Table 1 and Table 2). The first one pertained to general information about the farm. The second section addressed aspects related to farm management. Lastly, the third section focused on parasite infection and the use of anthelmintics. The responses to the survey facilitated the identification of high-risk management practices potentially linked to the development of AR, as previously described [11].

### 2.3. Animals and Faecal Samples

The collection of fecal samples and the administration of the survey were carried out simultaneously on the farms. On each farm, three pre-adult animals (6–12 months) and three adult animals (>12 months) were randomly selected for sampling. Fresh feces samples were individually collected from the rectum and immediately stored under refrigeration for transportation to the Parasitology Unit of the Department of Animal Health at the University of Córdoba, Spain. Upon arrival at the laboratory, samples were kept at 4 °C in a refrigeration chamber until coprological analyses were performed.

### 2.4. Faecal Examination

Before the analysis, two pools were created using fecal samples to obtain two samples per farm (one from three rearing animals and the other from three breeding animals). A fecal pool composed of three individual samples has shown no significant differences when evaluating the FEC compared to individual samples [27]. The parasite burden was expressed as eggs per gram of feces (EPG) or oocyst per gram of feces (OPG) using a modified McMaster technique [28]. Briefly, three grams of pooled feces were analyzed using a saturated zinc sulfate (Panreac^®^, Barcelona, Spain) solution (density = 1.35). Parasite’s eggs and oocysts were morphologically identified to the genus level (*Eimeria* spp., *Moniezia* spp., *Dicrocoelium dendriticum*, and *Trichuris* spp.) or, in the case of Strongyles, to suborder level (Strongylida). The identification of helminth eggs was performed according to MAFF [28].

### 2.5. Statistical Analysis

Survey responses were recorded into spreadsheets using Microsoft Excel. No statistical comparisons or repeatability evaluations were undertaken for this survey [29]. Data obtained from the different farms were expressed as frequencies and calculated separately for each farm for sheep and goats. For closed, multiple-choice questions, the results were expressed as the percentage of the selected answer.

Results obtained from coprology studies (OPG or EPG) were expressed as the cumulative parasite burden according to the species and age of the animals for the following gastrointestinal parasites: *Eimeria* spp., Strongyles, *Moniezia* spp., *Trichuris* spp., and *D. dendriticum*.

A farm was considered positive when one or more parasite egg/oocyst were detected by coprology. Prevalence was estimated for each gastrointestinal parasite species as the percentage of positive samples, and 95% confidence intervals (CI) were calculated using the exact binomial method [30]. The intensity of *Eimeria* spp. infection was categorized based on [31] as negative, low (<1800 OPG), medium (1800–6000 OPG), or high (>6000 OPG). Similarly, the intensity of Strongyles was classified as negative, low (<500 EPG), medium (500–1000 EPG), or high (>1000 EPG), according to Soulsby (1982).

Jamovi software v 2.3 was used to determine the risk factors associated with gastrointestinal parasite infections in sheep and goat flocks [32]. These factors were established according to the criteria proposed in [11]. Univariate and bivariate odds ratio analyses were performed. The analysis was based on a dichotomous outcome (flock positive or negative for each parasite species detected), and a herd was considered positive if at least one or more parasite eggs/oocysts were detected in the microscopical examination. A *p* value < 0.05 was considered statistically significant.

## 3. Results

### 3.1. Farm Questionnaire

The data collected in the surveys are shown in Table 1 and Table 2. For each question, the number of respondents was indicated in contrast to the total sample and expressed in percentage with 95% CI.

The farm and management characteristics of the surveyed sheep and goat farms are shown in Table 1. Sheep flocks had an average size of 599.83 (range 30–4064) and goat flocks of 309.82 animals (range 125–600). Sheep farms presented a similar percentage of flocks with less than 500 animals and more than 500 animals. Moreover, most goat farms (86.84%) had herds of fewer than 500 animals. The primary aptitude was meat production in sheep farms (92.45%) and exclusively dairy production in goat farms. Animal feed was principally based on forage (animals graze freely on the farm’s land), with concentrate in sheep (93.85%) and total mixed ration (TMR) in goats (94.59%). Regarding animal species, both sheep and goat farms were constituted mainly by one livestock species. Most surveyed farms had livestock as the only activity (79.49% sheep; 94.87% goat), although a small percentage combined livestock farming with agricultural activity (20.51% sheep; 5.13% goat). Most of the farms of both species reported that they did not graze the lambs/kids (90.48% sheep; 88.89% goat), although most of the goat farms practiced zero-grazing. However, most young grazers grazed with the rest of the flock of sheep farms compared with less than half of the goat farms.

Farmers’ perceptions about the use of anthelmintics and best management practices in sheep and goats are shown in Table 2. For both species, most farmers considered parasite infections an important issue (89.94% sheep; 92.31% goat), and the frequency of problems caused by parasites was classified as moderate. Interestingly, despite farmers’ perception, most did not carry out any coprological analyses in the last year and carried out deworming planning without veterinary advice, most through other means than their veterinarian (79.87% sheep; 76.92% goat). Regarding goat farmers, half of them had never asked for advice on parasite control in the last year. For both species, the main choices of anthelmintic drugs in the last year were macrocyclic lactones, followed by closantel and benzimidazole. It is noteworthy that some sheep farmers did not know the type of anthelmintic used during the last year. Despite the great variety of anthelmintic products, anthelmintic rotation was infrequent in sheep and goat farms. The frequency of drug treatments during the last year was mainly once a year or more, regardless of the age and the species. The most frequent deworming system performed in both species’ farms was based on applying it to the whole flock. Regarding the introduction of new animals, half of the sheep farmers dewormed, whereas this percentage was slightly higher in goat farmers. Farmers perceive access to contaminated pastures as a risk, especially after deworming. Thus, all avoided introducing animals to new pastures just after this practice. With respect to the presence of anthelmintic resistance, most did not observe anthelmintic resistance in their flocks (76.58% sheep; 94.74% goat).

The factors related to a hypothetical risk of developing AR are shown in Table 1 and Table 2. In relation to management systems, the answers seem to indicate a low risk of developing AR in both species. Nevertheless, sheep farms could present an intermediate risk in comparison with goat farms due to grazing new animals with the rest of the flock and the lack of deworming in new animals. Regarding parasite control and the use of anthelmintics, both species could present an intermediate risk based on the answers related to coprological analysis, veterinary advice, treatment frequency, and deworming practices. The lack of anthelmintic rotation entails a high risk that could lead to developing AR in both species. 

### 3.2. Cumulative Fecal Oocyst/Egg Counts

Cumulative fecal oocyst/egg counts (FOCs; FECs) are shown in Table 3. The highest counts were observed in coccidia infections. Specifically, the cumulative FOC of *Eimeria* spp. was lower in sheep compared to goats, although both species exhibited higher FOCs in pre-adult animals than in adults. The second most significant parasitic infection was caused by Strongyles. Consequently, the cumulative FEC in Strongyles was higher in sheep than in goats, with both species displaying lower FECs in pre-adults compared to adults. Parasitic infections caused by *Moniezia* spp. and *Trichuris* spp. were less prevalent but still showed higher cumulative FECs in sheep than in goats. Lastly, the presence of *D. dendriticum* was sporadic and only observed in adult sheep.

### 3.3. Prevalence and Infection Intensity of Coccidia and Gastrointestinal Helminths

As can be seen in Table 4, the percentage of samples identified as coccidia-negative was minimal in both species. Most adult sheep exhibited a medium parasite burden, while the highest burden was observed in pre-adult animals. Regarding strongyles, most sheep had a low parasite burden, whereas most goats tested were negative. In relation to *Moniezia* spp., *Trichuris* spp., and *D. dendriticum*, most animals were negative for the presence of these helminths. It is noteworthy that *D. dendriticum* was only detected in sheep.

### 3.4. Management System-Level Risk Factors Associated with the Presence of Strongyles

The odds ratio (OR) and 95% confidence interval (95% CI) were calculated for factors related to farm management practices and Strongyles infections in small ruminants. For sheep, meat production and a diet based on forage and concentrate were associated with the presence of parasites in the farm (OR = 9.31, CI = 1.93–44.9 and OR = 8.10, CI = 1.67–39.3, respectively), with a significant effect (*p* = 0.001 and *p* = 0.003 respectively) as indicated by the magnitudes of the odds ratios [33]. These results were corroborated in our study by the higher prevalence of Strongyles observed in pre-adult meat sheep (67.4%) compared to dairy ones (18.2%), as well as the higher prevalence seen in animals fed with forage and concentrate (64.3%) versus those fed a total mixed ration (TMR) (18.2%). No significant results were found for goats.

## 4. Discussion

### 4.1. Farm Questionnaire

To the author’s knowledge, to date, most surveys on farm characteristics, parasite control, and anthelmintic use in sheep and goat farms have been conducted primarily in northern Spain [17,18,20], with few studies performed in southern Spain [19]. However, this region is significant as it is home to more than half of the country’s small ruminant population [1]. In our survey, most sheep farms had a flock size of fewer than 500 animals. These results differed from those reported in previous studies [19], possibly due to the decrease in the sheep population in Spain from 2013 to the present day [1]. Conversely, the high proportion of goat herds with fewer than 500 animals found in our survey was consistent with previous studies carried out in southern Spain [34,35]. Additionally, the distribution based on aptitude (meat production in sheep: 92.5% and exclusively milk production in goats) has been previously reported [1,18]. As feeding type is associated with production modality, most sheep were fed with forage and concentrate, which is linked to extensive management [18,19,35], while the use of TMR is associated with dairy production, which has intensified in recent years [34,35]. Consistent with previous studies [18,35], the main activity of the farms surveyed in our study was dedicated exclusively to sheep, often with only a single species of livestock. It is well-known that GIN infections are associated with grazing [36], so not allowing newly weaned lambs to graze entails a protective factor against GIN infections. However, the percentage of farms that allowed new animals to graze with the rest of the flock in our study was surprisingly high (sheep 71.9%; goat 45.5%).

Parasitic infections were deemed significant by most surveyed farmers, who reported a moderate frequency of occurrence. However, despite farmers’ assertions, coprological analyses were only applied on rare occasions. This finding aligns with previous studies conducted in Spain [18,20] and Europe [24,25]. In our study, this was particularly relevant to goat farms.

The involvement of veterinarians in the deworming plan yielded mixed results, consistent with findings reported by other authors [18,20]. It has been demonstrated that veterinarians significantly influence farmers’ choice of anthelmintic drugs [18,19]. Regarding the frequency of treatment, it is interesting to note that in our study, biannual treatment was the primary option, as shown in previous studies in both southern [19] and northern Spain [17,18]. For goat farms, annual treatment was the most common frequency. This could be attributed to the increased intensification and reduced grazing associated with this type of livestock. Similar results have been reported in Northern Italy, where the most frequent treatment was once a year (73.6%) [25].

The anthelmintic choice was primarily macrocyclic lactones, followed by benzimidazoles, as reported in previous studies in southern Spain [18,19]. Macrocyclic lactones are the preferred choice in this region due to their persistent activity, better efficacy on GIN-inhibited larval stages, their effect on ectoparasites, and the convenience of using pour-on products [37]. In contrast, benzimidazoles were the primary choice in northern Spain and Italy [20,25]. Interestingly, 21.8% of sheep farmers were unsure about the drug they had used for treatments, while the opposite was true for goat farmers. As previously mentioned, this species-specific difference may be related to the intensive farming system employed for dairy goats. The percentage of farms (21.6–43.3%) that rotated drugs within the year aligns with previous studies in Spain [18,19]. 

In our survey, anthelmintics were typically administered to the entire herd. This practice often leads to dosage issues, as highlighted in several studies [18,25,37]. In this context, improper drenching practices could result in the under-dosing of anthelmintics. This could potentially lead to a lack of drug efficacy and a potential future problem of AR. This issue is particularly significant in goats due to their higher physiological tolerance and doses that are usually extrapolated from sheep [25,37]. Imported animals may serve as a significant entry route for GIN with AR [18,19,38,39]. In this study, the percentage of farms that dewormed imported animals was relatively low (50.4% sheep; 36.4% goats). This could be attributed to farmers’ belief that it is sufficient not to move these animals to pasture. Approximately 23% of the sheep farmers and only 5.2% of goat farmers reported suspecting the presence of AR. To our knowledge, there are no studies on AR in southern Spain. In Europe, the occurrence of AR in sheep and goats varied according to the geographic area and the drug used, ranging from 2.8 to 88% for benzimidazoles and 15.7 to 95% for macrocyclic lactones [17,21,22,23,25,26,40].

### 4.2. Cumulative Fecal Oocyst/Egg Counts

The higher cumulative FOC of *Eimeria* spp. in goats compared to sheep aligns with previous studies [41], and contrasts with other authors where the opposite situation was observed [24]. As anticipated, the development of robust immunity with aging resulted in a lower cumulative FOC observed in our study in adults compared to pre-adults in both species, which is consistent with previous studies [5,24,36]. Conversely, the higher cumulative Strongyles FECs in sheep than in goats contradicts previous studies that reported higher mean FECs in goats than in sheep due to a lower ability to elicit an effective immune response to nematodes [24,41]. Therefore, both the lower FEC and higher FOC observed in goats in the present study should be more closely associated with the intensive farming system and non-grazing practices [36,42]. Similarly, the lower FEC obtained in pre-adults compared to adult sheep could also be related to increased grazing in adult animals. The results obtained about cumulative FECs in *Moniezia* spp. and *D. dendriticum* (higher in sheep than goats) were consistent with previous reports [24,41].

### 4.3. Prevalence

The high prevalence (95–100%) of *Eimeria* spp. on sheep and goat farms agrees with a study conducted on pre-adult and adult sheep in Southern Spain [5], but contrasts with a lower prevalence recently observed in both sheep and goats in northern Spain [41]. This discrepancy between the two studies could be attributed to the varying climatic conditions in the two regions under study, which could significantly influence coccidia sporulation. The high intensity of *Eimeria* spp. infection in pre-adult animals compared to adult animals has been previously documented [24]. This could be due to the less developed immune system in younger animals, underscoring the importance of these parasites for lambs and kids [36].

The prevalence of Strongyles observed in our study on sheep farms (63–66%) was similar to the only national study available for comparison [41], but is lower compared to the results found in Northern Italy (68–84%) [24]. Regarding goats, the prevalence of Strongyles (3–33%) found in our study was lower than reported in previous studies performed in Spain and Italy, which indicated a prevalence of 73–84% [24,41]. Our findings are also consistent with earlier studies performed on goats in northern Italy [36,42,43]. The discrepancy between these studies could be explained by factors such as the grazing period, climatic conditions in the areas under study, and the application of anthelmintic treatments. The differences in prevalence between the two species could be due to divergent evolutionary processes, such as feeding behavior and immune response, in relation to GIN infections [7]. Additionally, in this study, the main source of goat feed was total mixed ration TMR. Therefore, the practice of non-grazing reduced the likelihood of infection. For the same reason, the intensity of infection was higher in sheep than in goats, as Strongyles infection is associated with grazing [36,42].

The prevalence of *Moniezia* spp. in our study was higher in sheep than in goats, which aligns with previous studies performed in Spain and Poland [41,44]. However, the prevalence values obtained in our study exceeded those reported by the aforementioned authors. The exception was one study in which a prevalence in sheep of 53% in non-irrigated pasture compared to irrigated pasture (9.5%) was found [45]. In terms of goats, other studies carried out in northern Italy showed a higher prevalence than our study [42,43], while studies conducted in Poland [44] and northern Spain [41] reported a lower prevalence. The prevalence of *Trichuris* spp. in both ruminant species was lower than that described in other studies in northern Spain [45,46] and northern Italy [24,25,36,42,43]. Regarding *D. dendriticum*, our results are similar to those reported in northern Italy [24] and higher than previously reported in northern Spain [41]. This discrepancy in prevalence could be attributed to differences in temperature and the abundance of naturally irrigated pastures [45,47].

### 4.4. Risk Factors in Farm Management Practices Associated with Gastrointestinal Strongyles Infection

In our study, the risk factors associated with gastrointestinal strongyle infection were identified as aptitude and feeding. Most of the sheep farmers surveyed indicated forage and concentrate as the main feeding system. This agrees with the association found between Strongyles infection and a feed system based on forage and concentrate in pre-adult sheep, as the free-living stages of Strongyles find a suitable environment for growth on pasture [36,42,43]. Therefore, we believe that the association of meat aptitude with the presence of strongyles in pre-adult sheep could be associated more with the feeding system than with aptitude itself. In addition, in Spain, sheep meat farms typically operate on extensive systems and utilize local breeds, which are known for their resilience and easy adaptation to the harsh weather conditions of these regions [35,48,49]. 

## 5. Conclusions

In our study, the risk factors associated with gastrointestinal strongyle infection were identified as aptitude and feeding. Most of the sheep farmers surveyed indicated forage and concentrate as the main feeding system. This agrees with the association found between Strongyles infection and a feed system based on forage and concentrate in pre-adult sheep, as the free-living stages of Strongyles find a suitable environment for growth on pasture [36,42,43]. Therefore, we believe that the association of meat aptitude with the presence of strongyles in pre-adult sheep could be associated more with the feeding system than with aptitude itself. In addition, in Spain, meat sheep farms typically operate on extensive systems and utilize local breeds, which are known for their resilience and easy adaptation to the harsh weather conditions of these regions [35,48,49]. 

## Figures and Tables

**Table 1 animals-14-01668-t001:** Farm characteristics and pasture management strategies in the sheep and goat farms surveyed.

	Sheep (*n* = 159 Flocks)		Goat (*n* = 39 Herds)		Risk of AR Development ^2^
	*n* ^1^	%	*n*	%	
**Flock/Herd Size (Animals/Herd)**	150		38		
<500	72	48.00 (39.78–56.30)	33	86.84 (71.91–95.59)	**-**
500–1000	55	36.67 (28.96–44.92)	5	13.16 (4.41–28.09)	**-**
>1000	23	15.33 (9.97–22.11)	0	0.00	**-**
**Aptitude**	159		39		
Meat	147	92.45 (87.19–96.04)	0	0.00	**-**
Dairy	12	7.55 (3.96–12.80)	39	100.00	**-**
**Feed ^3^**	130		37		
Total mixed ration	12	9.23 (4.86–15.57)	35	94.59 (81.80–99.34)	**-**
Forage and concentrate	122	93.85 (88.23–97.31)	8	21.62 (9.83–38.21)	**-**
**Number of species in the farm**	156		39		
Single specie	94	60.26 (52.12–67.99)	34	87.18 (72.57–95.70)	**-**
More than one species	62	39.74 (32.01–47.88)	5	12.82 (4.30–27.43)	**-**
**Farm activity**	156		39		
Only Livestock	124	79.49 (72.29–85.53)	37	94.87 (82.68–99.37)	**-**
Livestock and agricultural	32	20.51 (14.47–27.71)	2	5.13 (0.63–17.32)	**-**
**After weaning. do you move lambs to pasture?**	147		9		
Yes	14	9.52 (5.31–15.46)	1	11.11 (0.28–48.25)	**Intermediate**
No	133	90.48 (84.54–94.69)	8	88.89 (51.75–99.72)	**Low**
**Do the newly introduced sheep graze with the rest of the flock?**	135		11		
Yes	97	71.85 (63.47–79.25)	5	45.45 (16.75–76.62)	**Intermediate**
No	38	28.15 (20.75–36.53)	6	54.55 (23.38–83.25)	**Low**

95% CI is expressed in brackets. ^1^ Numbers in the first line of every question indicate the number of answers to this question. *n* may be lower than the total number of farms surveyed due to unanswered or non-applicable questions. ^2^ Estimation of risk to develop anthelmintic resistance according to factors reported by [11]. ^3^ Multiple choice question.

**Table 2 animals-14-01668-t002:** Farmers’ perceptions on parasitic infections, anthelmintic uses, and best management strategies in the sheep and goat farms surveyed.

	Sheep (*n* = 159 Flocks)		Goat (*n* = 39 Herds)		Risk of AR Development ^2^
	*n* ^1^	%	*n*	%	
**Do you consider problems caused by parasites to be important?**	159		39		
Yes	143	89.94 (84.17–94.14)	36	92.31 (79.13–98.38)	
No	9	5.66 (2.62–10.47)	1	2.56 (0.06–13.48)	
I don’t know	7	4.40 (1.79–8.86)	2	5.13 (0.06–17.32)	
**What is the frequency of problems caused by parasites in your herd?**	159		39		
Low	62	38.99 (31.37–47.04)	10	25.64 (13.04–42.13)	
Moderate	90	56.60 (48.52–64.43)	28	71.79 (55.13–85)	
High	7	4.40 (1.79–8.86)	1	2.56 (0.06–13.48)	
**Do you perform at least one coprological analysis per year?**	158		39		
No	150	94.94 (90.27–97.79)	34	87.18 (72.57–95.70)	Intermediate
Yes	8	5.06 (2.21–9.73)	5	12.82 (4.30–27.43)	Low
**Is the veterinarian involved in planning deworming?**	159		39		
Yes	32	20.13 (14.19–27.21)	9	23.08 (11.13–39.33)	Low
No	127	79.87 (72.79–85.81)	30	76.92 (60.67–88.87)	Intermediate
**What has been the frequency of parasite control counseling in the last year?**	155		39		
Never	42	27.10 (20.28–34.81)	20	51.28 (34.78–67.58)	High
Once a year	65	41.94 (34.07–50.12)	15	38.46 (23.36–55.38)	Intermediate
Twice a year	35	22.58 (16.26–29.98)	4	10.26 (2.87–24.22)	Intermediate
More than twice a year	13	8.39 (4.54–13.92)	0	0.00	Low
**What has been the frequency of treatment of adult animals in the last year?**	150		37		
Never	2	1.33 (0.16–4.73)	1	2.70 (0.07–14.16)	Low
Once a year	59	39.33 (31.47–47.63)	32	86.49 (71.23–95.46)	Intermediate
Twice a year	82	54.67 (46.34–62.80)	4	10.81 (3.03–25.42)	Intermediate
More than twice a year	7	4.67 (1.90–9.38)	0	0.00	High
**What has been the frequency of treatment of pre-adult animals in the last year?**	127		38		
Never	4	3.15 (0.86–7.87)	2	5.26 (0.64–17.75)	Low
Once a year	62	48.82 (39.85–87.84)	26	68.42 (51.35–82.50)	Intermediate
Twice a year	60	47.24 (38.32–56.30)	10	26.32 (13.40–43.10)	Intermediate
More than twice a year	1	0.79 (0.02–4.31)	0	0.00	High
**What has been the frequency of treatment of young animals in the last year?**	93		36		
Never	13	13.98 (7.66–22.72)	5	13.89 (4.67–29.50)	Low
Once a year	70	75.27 (65.24–83.63)	23	63.89 (46.22–79.18)	Intermediate
Twice a year	7	7.53 (3.08–14.89)	7	19.44 (8.19–36.02)	Intermediate
More than twice a year	3	3.23 (0.67–9.14)	1	2.78 (0.07–14.53)	High
**What kind of dewormers have you used in the last year? ^3^**	156		38		
I don’t know	34	21.79 (15.59–29.10)	0	0.00	-
None	2	1.28 (0.16–4.55)	1	2.63 (0.07–13.81)	-
Benzimidazole	45	28.85 (21.88–36.63)	20	52.63 (35.82–69.02)	-
Macrocyclic lactones	85	54.49 (46.33–62.47)	23	60.53 (43.39–75.96)	-
Clorsulon	1	0.64 (0.02–3.52)	1	2.63 (0.07–13.81)	-
Closantel	46	29.49 (22.46–37.31)	1	2.63 (0.07–13.81)	-
Levamisole	5	3.21 (1.05–7.32)	0	0.00	-
Nitroxinil	2	2.35 (0.16–4.55)	0	0.00	-
Other	0	0.00	0	0.00	-
**In case of deworming. drug rotation**	120		37		
Yes	52	43.33 (34.32–52.69)	8	21.62 (9.83–38.21)	Low
No	68	56.67 (47.31–65.68)	29	78.38 (61.79–90.17)	High
**Application of deworming**	157		38		
Whole flock	154	98.09 (94.52–99.60)	38	100.00	Intermediate
Individually	3	1.91 (0.40–5.48)	0	0.00	Low
**Deworming in imported animals**	127		11		
Yes	63	49.61 (40.62–58.61)	7	63.64 (30.79–89.07)	Low
No	64	50.39 (41.39–59.38)	4	36.36 (10.93–69.21)	Intermediate
**Immediately after deworming. move animals to new pastures**	150		9		
No	145	96.67 (92.39–98.91)	9	100.00	Low
Yes	5	3.33 (1.09–7.61)	0	0.00	Intermediate
**Do you suspect the presence of anthelmintic resistance in your herd?**	158		38		
Yes	36	22.78 (16.50–30.12)	2	5.26 (0.64–17.75)	-
No	121	76.58 (69.20–82.94)	36	94.74 (82.25–99.36)	-
I don’t know	1	0.63 (0.02–3.48)	0	0.00	-

95% CI is expressed in brackets. ^1^ Numbers in the first line of every question indicate the number of answers to this question. *n* may be lower than the total number of farms surveyed due to unanswered or non-applicable questions. ^2^ Estimation of risk to develop anthelmintic resistance according to factors reported by [11]. ^3^ Multiple choice question.

**Table 3 animals-14-01668-t003:** Results from a coprological study using the feces pool from three pre-adult animals (6–12 months) and three adult animals (>12 months) sampled in each farm (expressed as cumulative * opg/hpg in all studied farms).

	Sheep		Goat	
	Pre-Adult (4–12 Months)	Adult (>12 Months)	Pre-Adult (4–12 Months)	Adult (>12 Months)
***Eimeria* spp. (OPG)**	897,450 (0–64,500)	108,950 (0–8150)	1,338,600 (150–258,500)	125,600 (0–11,050)
**Strongyles (EPG)**	17,200 (0–900)	26,150 (0–2900)	350 (0–350)	1750 (0–400)
***Moniezia* spp. (EPG)**	9550 (0–1900)	1850 (0–250)	50 (0–50)	100 (0–50)
***Trichuris* spp. (EPG)**	600 (0–150)	50 (0–50)	0 (-)	50 (0–50)
***D. dendriticum* (EPG)**	0 (-)	400 (0–150)	0 (-)	0 (-)

* results are expressed as the sum of the OPG or EPG of all farms. Range is expressed in brackets.

**Table 4 animals-14-01668-t004:** Results (prevalence) of parasite infection in the sheep and goat farms surveyed.

	Sheep				Goat			
	Pre-Adult (*n* = 146)		Adult (*n* = 155)		Pre-Adult (*n* = 39)		Adult (*n* = 39)	
	*n* ^1^	%	*n*	%	*n*	%	*n*	%
***Eimeria* spp.**								
Negative	2	1.37 (0.17–4.86)	8	5.16 (2.25–9.92)	0	0.00	1	2.56 (0.06–13.48)
Low (<1800 OPG)	94	64.38 (56.04–72.13)	4	2.58 (0.71–6.48)	1	2.56 (0.06–13.48)	8	20.51 (9.30–36.46)
Medium (1800–6000 OPG)	9	6.16 (2.86–11.38)	142	91.61 (86.08–95.46)	9	23.08 (11.13–39.33)	27	69.23 (52.43–82.98)
High (>6000 OPG)	41	28.08 (20.97–36.11)	1	0.65 (0.02–3.54)	29	74.36 (57.87–86-96)	3	7.69 (1.62–20.87)
**Strongyles**								
Negative	54	36.99 (29.15–45.36)	53	34.19 (26.77–42.23)	38	97.44 (86.52–99.94)	26	66.67 (49.78–80.91)
Low (<500 EPG)	86	58.90 (50.47–66.97)	90	58.06 (49.88–65.93)	1	2.56 (0.06–13.48)	13	33.33 (19.09–50.22)
Medium (500–1000 EPG)	6	4.11 (1.52–8.73)	9	5.81 (2.69–10.74)	0	0.00	0	0.00
High (>1000 EPG)	0	0.00	3	1.94 (0.4–5.55)	0	0.00	0	0.00
***Moniezia* spp.**								
Negative	109	74.66 (66.80–81.49)	131	84.52 (77.84–89.82)	38	97.44 (86.52–99.94)	37	94.87 (82.68–99.37)
Positive	37	25.34 (18.51–33.20)	24	15.48 (10.18–22.16)	1	2.56 (0.06–13.48)	2	5.13 (0.63–17.32)
***Trichuris* spp.**								
Negative	137	93.84 (88.62–97.14)	154	99.35 (96.46–99.98)	39	100.00	38	97.44 (86.52–99.94)
Positive	9	6.16 (2.86–11.38)	1	0.65 (0.02–3.54)	0	0.00	1	2.56 (0.06–13.48)
** *D. dendriticum* **								
Negative	146	100.00	149	96.13 (91.76–98.57)	39	100.00	39	100.00
Positive	0	0.00	6	3.87 (1.43–8.24)	0	0.00	0	0.00

95% CI are expressed in brackets. ^1^ Number of samples. *n* may be lower than the total number of farms surveyed because fecal samples could not be collected in all farms.

## Data Availability

The dataset is available upon request from the authors.

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
