# Peer review of "Prevalence of Gastrointestinal Parasites in Small Ruminant Farms in Southern Spain"

_animals, 2024, doi:10.3390/ani14111668_

Round 1
Reviewer 1 Report
The manuscript has the strength of gathering questionnaire results of large number of sheep flocks and goat herds comprising large number of animals from a defined geographical area from Southern Spain. On the other hand, prevalence was done with faecal samples collected in two months, which is considered a faily short period of time.
My suggestion is to consider this manuscript for publication. The discussion was thoroughly written and it deserves to be published. Few comments are as follows:
Line 100: Include total numbers, plus min and max n/flock in the sheep study and numbers plus min and max n/herd for goats examined during this study; It is mentioned in lines 164-165 but it could be quickly mentioned in M&M;
Line 101: "April 2021 to May 2021" is that correct? Please justify such short prevalence study;
Line 168: Would be there a way to gather sheep flocks and goat herds by nutritional supplementation in order to highlight differences in parasite genera? The same for Line 173 with zero grazing goat farms?
Line 174: Perhaps the term "young grazers" could replace "new animals"?
Line 177: It is interesting to note that "most farmers considered parasite infections an important issue". It looks like these animals do not have parasite problems at all;
Line 179: How far are the parasitological labs around this geographical area and how long the labs take to get results back to farmers? Sometimes logistics related to sending faecal samples are hard to stablish thus the farmers are not to be blamed. However, if farmers would drench their animals regardless lab results, then this is another story;
Line 181: Would that be zero-grazing goats?
Line 193: Farmers did not perceive AH resistance? FEC reduction tests were ran?
Table 1: "Forage and concentrate" for goat herds should not be lower than 21.62%;
Table 2: Should "What has been the frequency of treatment of young animals in the last year?" and "What kind of dewormers have you used in the last year?" should these two questions not be crossed to assess AH resistance risk?
Table 3: Eimeria spp results are higher in goat herds and compatible with indoor production systems. The same is true for Strongyles, Moniezia, Trichuris and D. dendriticum in pre-adult goats. What was the percentage of indoor sheep flocks? It seems there should be a high % of sheep being reared indoors;
Lines 252-253: 67.4% and 18.2% where?
Line 352: what does TRM stand for?
Line 383: Could "aptitude" be confounded with rearing system? Indoor/outdoor systems?
Author Response
- “Line 100: Include total numbers, plus min and max n/flock in the sheep study and numbers plus min and max n/herd for goats examined during this study; It is mentioned in lines 164-165 but it could be quickly mentioned in M&M.”
Response: Thank you very much for these observations. The changes have been included in the manuscript as suggested.
- “Line 101: "April 2021 to May 2021" is that correct? Please justify such short prevalence study;”
Response: This is correct. The reason for this brief prevalence study is because during this period, the newly introduced animals transition to the next phase of production and cohabitated with the adult animals. This is why the samples were categorized into two groups: “young animals” (weaning) and “adults”. Conversely, the period from April to May constitutes spring in the southern region of Spain, during which the prevalence of gastrointestinal parasites could potentially reach its peak.
- “Line 168: Would be there a way to gather sheep flocks and goat herds by nutritional supplementation in order to highlight differences in parasite genera? The same for Line 173 with zero grazing goat farms?”
Response: Thank you so much for this insightful observation. Indeed, it could be interesting to highlight differences in parasites genera. In this study, our primary objective was to know the data of prevalence of gastrointestinal parasites, given the lack of such studies conducted in southern Spain. For futures research, this pertinent observation will be considered in conjunction with a revised sampling design.
- “Line 174: Perhaps the term "young grazers" could replace "new animals"?”
Response: Thank you very much for these observations. The changes have been included in the manuscript as suggested.
- “Line 177: It is interesting to note that "most farmers considered parasite infections an important issue". It looks like these animals do not have parasite problems at all.”
Response: In this sentence, we aim to highlight that farmers perceive parasitic infections as a significant concern, to the extent that they categorize their “parasitic issues” as moderate (line 190). However, the survey responses reveal a contradiction. Farmers sometimes administer treatments to the animals themselves, often without knowing the principles of the medication used, or the neglect to seek veterinary advise for deworming. This discrepancy is noteworthy; they are aware of the problems caused by parasites, yet they do not consistently implement appropriate management strategies to control them.
- “Line 179: How far are the parasitological labs around this geographical area and how long the labs take to get results back to farmers? Sometimes logistics related to sending faecal samples are hard to stablish thus the farmers are not to be blamed. However, if farmers would drench their animals regardless lab results, then this is another story;”
Response: The parasitological lab is located within the Animal Health Department (Parasitology Unit) of the Veterinary Faculty of Córdoba (about two hours by car from COVAP). The time for delivering the analysis results to farmers was one week. This represents a significant effort for the Parasitology Unit, as despite the number of samples were low, the final volume of samples was quite high. In the lab, we receive a box of faecal samples weekly (usually on Wednesdays), with six samples per farm (three for young animals and three for adults). This implied that samples were collected on Tuesday morning, then packed and sent on Wednesday morning. The samples were received in the lab the same day, typically around 12:00 or 13:00 h. As previously mentioned, the primary objective of this study was to determine the prevalence of gastrointestinal parasites. At this juncture, we must acknowledge the farmers and, in particular, the Research and Development Department of COVAP, whose cooperation made it possible to implement the logistical management of the samples for this study.
- “Line 181: Would that be zero-grazing goats?”
Response: This is true. Goat farms were all of them indoor because were dairy goats. The only animals that graze were the sheep.
- “Line 193: Farmers did not perceive AH resistance? FEC reduction tests were ran?”
Response: This is true. Goat farms were all of them indoor because were dairy goats. The only animals that graze were the sheep.
- “Table 1: "Forage and concentrate" for goat herds should not be lower than 21.62%;”
Response: The data presented in brackets represent the 95% confidence interval. This value is derived using statistical software. A footnote at the end of Table 1 provides an explanation for this annotation.
- “Table 2: Should "What has been the frequency of treatment of young animals in the last year?" and "What kind of dewormers have you used in the last year?" should these two questions not be crossed to assess AH resistance risk?”
Response: Undoubtedly, these two questions could be crossed. However, we have chosen to keep them separate in order to apply the same risk criteria for Anthelmintic Resistance (AR) as outline Silvestre et al., 2002. By doing so, we were able to pose questions similar to those of the aforementioned author and subsequently compare our results.
- “Table 3: Eimeria spp results are higher in goat herds and compatible with indoor production systems. The same is true for Strongyles, Moniezia, Trichuris and D. dendriticum in pre-adult goats. What was the percentage of indoor sheep flocks? It seems there should be a high % of sheep being reared indoors;”
Response: This question was not explicitly formulated. In Spain, the sheep production system involves an indoor setup for pre-adult animals. Once these animals reach sexual maturity, they are integrated with adult animals and transition to grazing alongside them. During the pre-adult stage, they are more susceptible to conditions such as coccidiosis and certain parasites (such as Moniezia). Additionally, the production system is influenced by the intended purpose: meat production is closely associated with grazing, while dairy production tends to favour indoor system.
- “Lines 252-253: 67.4% and 18.2% where?”
Response: This statement refers to our study. The necessary corrections have been made in the manuscript.
- “Line 352: what does TRM stand for?”
Response: The term “TRM” refers to Total Mixed Ration. The necessary corrections have been made in the manuscript.
- “Line 383: Could "aptitude" be confounded with rearing system? Indoor/outdoor systems?”
Response: When we employ the term “aptitude” we are referring to the intended use of production, whether it be for meat of dairy.

Reviewer 2 Report
Comments and Suggestions for Authors
This is an interesting article which tests small numbers of small ruminant faeces for gastrointestinal parasites on farms across southern Spain. Although small numbers per farm are used, there are a large number of farms tested which adds to the weight of the paper.
I think that the sample numbers are adequate and the testing carried out is good, along with some nice analysis. The questionnaire is nicely executed, but I feel it would benefit from being written up with more of the data included rather than ‘most respondents …..’
Also perhaps a good proof read just to polish the paper would be useful
I have some more specific comments below
Please ensure that the references are done correctly, i.e. in square brackets
Is it worth a mention of the abortive potential of parasites such as Neospora?
Line 81- please reword as this doesn’t make sense
Questionnaire- would it be possible to include the questionnaire as supplementary data, This may then allow people to repeat the study in different parts of the world allowing for an exact comparison?
Line 117- how was this done randomly?
Line 142- how were these parasites chosen? Were they the only ones seen?
Line 153- wrong referencing style
Section 3.1. I appreciate that the tables are there, but this is all a bit vague. It would be nice to include some values rather than most, or some just for clarity for the reader
Table 1 and 2- please do not leave blank boxes as it makes it look like data is missing. Can you please add a hyphen or even merge the boxes and grey them out etc?
Section 3.2. please check italicisation of the Latin names
Table 3- would the introduction of a range here be useful for understanding the distribution of the parasites
Line 236- most adult sheep … reword
Line 259- to the authors knowledge. … reword
Line 320- incorrect referencing style
Line 328- could this just be a time thing that younger animals haven’t had the time to be exposed to many parasites yet?
Section 4.3. Is there any genetic differences- i.e. different breeds- between northern and southern spain? Just a thought
Line 359- incorrect referencing style
Comments on the Quality of English Language
These are detailed above. A few minor points where it doesnt make complete sense, but nothing that a good proof read wont solve
Author Response
Reviewer 2
Comments and Suggestions for Authors
“This is an interesting article which tests small numbers of small ruminant faeces for gastrointestinal parasites on farms across southern Spain. Although small numbers per farm are used, there are a large number of farms tested which adds to the weight of the paper.
I think that the sample numbers are adequate and the testing carried out is good, along with some nice analysis. The questionnaire is nicely executed, but I feel it would benefit from being written up with more of the data included rather than ‘most respondents …..’
Also perhaps a good proof read just to polish the paper would be useful
I have some more specific comments below”
•“Please ensure that the references are done correctly, i.e. in square brackets”
Response: The necessary corrections have been made in the manuscript.
•“Is it worth a mention of the abortive potential of parasites such as Neospora?”
Response: Although Neospora can be an important parasite to consider, the primary objective of this study was to determine the prevalence data of gastrointestinal parasites and identify risk management practices that could potentially impact their presence in flocks. Nevertheless, this insightful observation will be taken into account for futures studies.
•“Line 81- please reword as this doesn’t make sense”
Response: The statement has been revised for improved clarity and ease of comprehension.
•“Questionnaire- would it be possible to include the questionnaire as supplementary data, This may then allow people to repeat the study in different parts of the world allowing for an exact comparison?”
Response: Of course, the questionnaire will be included as supplementary data.
•“Line 117- how was this done randomly?”
Response: The animals were randomly selected through a collaborative process involving our colleagues from the Research and Development Department of COVAP and the farmers. Upon the arrival of new animals (pre-adults) to the flock, three of them were chosen for faecal sampling. The same selection method was applied to the adult animals.
•“Line 142- how were these parasites chosen? Were they the only ones seen?”
Response: The parasite observed were not selectively chosen. As previously mentioned, the primary objective of this study was to determine the prevalence data of gastrointestinal parasites. Consequently, the genera and species commented were identified through coprological analysis.
•“Line 153- wrong referencing style”
Response: The necessary corrections have been made in the manuscript.
•“Section 3.1. I appreciate that the tables are there, but this is all a bit vague. It would be nice to include some values rather than most, or some just for clarity for the reader”
Response: The necessary corrections have been made in the manuscript.
•“Table 1 and 2- please do not leave blank boxes as it makes it look like data is missing. Can you please add a hyphen or even merge the boxes and grey them out etc?”
Response: The necessary corrections have been made in the manuscript.
•“Section 3.2. please check italicisation of the Latin names”
Response: The necessary corrections have been made in the manuscript.
•“Table 3- would the introduction of a range here be useful for understanding the distribution of the parasites”
Response: The necessary corrections have been made in the manuscript.
•“Line 236- most adult sheep … reword”
Response: The necessary corrections have been made in the manuscript.
•“Line 259- to the authors knowledge. … reword”
Response: The necessary corrections have been made in the manuscript.
•“Line 320- incorrect referencing style”
Response: The necessary corrections have been made in the manuscript.
•“Line 328- could this just be a time thing that younger animals haven’t had the time to be exposed to many parasites yet?”
Response: In our view, this is associated with the production system. All the goat farms were focused on dairy production; therefore, the animals were kept in an intensive indoor system. This environment is more conductive to the development of coccidiosis rather than other parasites.
•“Section 4.3. Is there any genetic differences- i.e. different breeds- between northern and southern spain? Just a thought”
Response: Thank you very much for your observation; it raises an intriguing point. Although there are distinct breeds across northern and southern Spain, the animals are frequently affected by identical parasites. The primary distinction may lie in the environmental conditions, which tend to be more favorable in the north and more restrictive in the south. Recent studies have concentrated on the resistance of certain breeds to various parasites.
•“Line 359- incorrect referencing style”
Response: The necessary corrections have been made in the manuscript.
